# Systematic review of process evaluations of interventions in trials investigating sedentary behaviour in adults

Jessica Faye Johansson [1,2] Natalie Lam,[1,2] Seline Ozer,[1,2] Jennifer Hall,[2,3]
Sarah Morton,[4] Coralie English [5] Claire F Fitzsimons,[6] Rebecca Lawton [7,8]
Anne Forster,[1,2] David Clarke [1,2]

For numbered affiliations see end of article.

**Correspondence to**
Dr Jessica Faye Johansson;
jessica.johansson@bthft.nhs.uk

## ABSTRACT

**Objectives** To systematically review and synthesise findings from process evaluations of interventions in trials which measured sedentary behaviour as an outcome in adults to explore: (1) how intervention content, implementation, mechanisms of impact and context influence outcomes and (2) how these interventions are experienced from different perspectives (participants, carers, staff).

**Design** Systematic review and narrative synthesis underpinned by the Medical Research Council process evaluation framework.

**Data sources** Databases searches were conducted in March 2019 then updated in May 2020 and October 2021 in: CINAHL, SPORTDiscus, Cochrane Database of Systematic Reviews, Cochrane Central Register of Controlled Trials, AMED; EMBASE, PsycINFO, MEDLINE, Web of Science and ProQuest Dissertations & Theses.

**Eligibility criteria** We included: Process evaluations of trials including interventions where sedentary behaviour was measured as an outcome in adults aged 16 or over from clinical or non-clinical populations. We excluded studies if interventions were delivered in educational or workplace settings, or if they were laboratory studies focused on immediate effects of breaking sitting.

**Data extraction and synthesis** Two independent reviewers extracted and coded data into a framework and assessed the quality of studies using the Mixed Methods Appraisal Tool. We synthesised findings using a narrative approach.

**Results** 17 process evaluations were included. Five interventions focused on reducing sedentary behaviour or sitting time, 12 aimed to increase physical activity or promote healthier lifestyles. Process evaluations indicated changes in sedentary behaviour outcomes were shaped by numerous factors including: barriers (eg, staffing difficulties and scheduling problems) and facilitators (eg, allowing for flexibility) to intervention delivery; contextual factors (eg, usual lifestyle and religious events) and individual factors (eg, pain, tiredness, illness, age and individual preferences).

**Discussion** Intervention requires careful consideration of different factors that could influence changes in sedentary behaviour outcomes to ensure that interventions can be tailored to suit different individuals and groups.

**PROSPERO registration number** CRD42018087403.

## Strengths and limitations of this study

► This systematic review is guided by Preferred Reporting Items for Systematic Reviews and Meta-analyses guidance.
► This is the first systematic review which has synthesised data from process evaluations evaluating interventions in trials that measure sedentary behaviour as an outcome in adults.
► The Medical Research Council guidance for conducting process evaluations has been used to structure this review and provided a comprehensive way of identifying factors associated with implementation, mechanisms of impact and context which may influence the effectiveness of randomised controlled trials investigating sedentary behaviour in adults.
► Non-English electronic databases were not searched. This limitation may cause language bias.
► There is some inconsistency in the quality of the reporting of the process evaluations included in the review.

## INTRODUCTION

Sedentary behaviour is defined as any waking behaviour characterised by energy expenditure ≤1.5 Metabolic Equivalents while in a sitting, lying or reclining posture.[1] In recent years, research exploring sedentary behaviour in adults has been expanding rapidly, documenting the potential for sedentary behaviour to have detrimental effects on health, well-being, and healthcare costs.[2] Randomised controlled trials (RCTs) are particularly useful to examine intervention effectiveness.[3] However, this approach cannot fully account for how interventions work, and the degree to which intervention components contribute to effectiveness or ineffectiveness.[4]

Interventions targeting sedentary behaviour are typically complex, with multiple interacting components.[5] Changes in outcomes following interventions are largely influenced

**BMJ**

Box 1   Review objectives

1. To identify and record the trial data (eg, design of interventions, sample sizes, duration and content of interventions, and primary and secondary outcome data (from the process evaluation publication or associated publications).
2. Establish whether logic models or theoretical models were used to explain how interventions were intended to work.
3. Establish whether interventions were delivered as intended (as per protocol).
4. Explore intended or unintended mechanisms that influence the extent to which interventions are effective.
5. Understand barriers and facilitators to delivery of, and participation in, interventions and any recommendations made to address such barriers and facilitators.
6. To synthesise qualitative data concerning the understanding and experiences of interventions from the perspectives of participants, family members/carers and intervention staff.

by human behaviours and contextual factors as part of a complex process.[6] The value of studying intervention processes, is recognised in the Medical Research Council (MRC) guidelines for developing and evaluating complex interventions[3] and detailed in the guidance for conducting process evaluations of complex interventions.[4] Process evaluations are designed to help understand the theoretical assumptions underpinning an intervention, and to disentangle factors which may have contributed to the outcomes of an intervention.[4]

The MRC process evaluation framework states that understanding of causal assumptions underpinning interventions and evaluation of how interventions work in practice are vital in building an evidence base that informs policy and practice. The framework outlines key functions of a process evaluation including investigating implementation, mechanisms of impact and context to understand how outcomes are interpreted.[4]

To date, systematic reviews have synthesised the evidence of effectiveness of interventions aimed at reducing sedentary behaviour.[7 8] However, it is also important to synthesise findings from process evaluations to understand the complexity of factors that may influence whether interventions are effective in reducing sedentary behaviour as these will inform future interventions in this relatively new research area. This paper seeks to address the following aims and objectives (box 1).

## Aims and objectives

1. To identify process evaluations of interventions in trials which measured sedentary behaviour as an outcome in adults, to understand the intervention content, mechanisms of impact, implementation and delivery approaches and contexts, in which interventions were reported to be effective or ineffective.
2. To explore experiences of participants, family members/carers and intervention staff in interventions that measured sedentary behaviour as an outcome in adults.

Qualitative data related to exploring perceptions, views and lived experiences of sedentary behaviour, but not related to receipt or delivery of an intervention were examined in a separate systematic review.[9]

The MRC process evaluation framework[4] was the underpinning framework for this review informing the aims and objectives, coding framework, providing a structure for synthesising and reporting findings.

## METHODS
### Protocol and registration
Reporting of this systematic review is guided by Preferred Reporting Items for Systematic Reviews and Meta-analyses (PRISMA) guidance (online supplemental file 1).[10 11]

### Patient and public involvement
No patients involved.

### Eligibility criteria
#### Study design
Studies explicitly identified by authors as a process evaluation, or studies that aimed to understand the functioning of an intervention by examining implementation, mechanisms of impact, and contextual factors.[12] Only process evaluations of RCTs, cluster RCTs, and randomised cross-over trials were included. Cohort and uncontrolled before-and-after studies were excluded.

#### Participants
Adults aged 16 or over regardless of whether they were recruited from a clinical or nonclinical population.

#### Interventions
Interventions which measured sedentary behaviour as an outcome, even if reducing sedentary behaviour was not the primary outcome.

Interventions were excluded if: they were delivered primarily in schools, colleges, universities or a workplace; or aimed at the acute (immediate) effects of breaking up sitting time as part of a supervised (usually laboratory-based) intervention.

#### Comparators
In trials, intervention groups may be compared with: no treatment, usual care, attention control, waitlist control groups or alternative treatments.

### Information sources
#### Electronic sources
In collaboration with information specialist colleagues, comprehensive search strategies were developed using controlled vocabulary and free-text terms (online supplemental file 2 for the search strategy for the MEDLINE database). Searches were conducted in March 2019 then updated in May 2020 and October 2021.

We searched the following databases: CINAHL (EBSCOHost), SPORTDiscus (EBSCOHost), Cochrane Database of Systematic Reviews (Wiley), Cochrane Central Register

of Controlled Trials (Wiley), AMED (OVID); EMBASE (OVID), PsycINFO (OVID), Ovid MEDLINE(R), OVID MEDLINE(R) and Epub Ahead of Print, In-Process & Other Non-Indexed Citations, Web of Science: Sciences Citation Index Expanded (Clarivate), Web of Science: Social Sciences Citation Index Expanded (Clarivate), Web of Science: Conference Proceedings Citation Index-Science (Clarivate), Web of Science: Conference Proceedings Citation Index-Social Sciences and Humanities (Clarivate), ProQuest Dissertations & Theses.

### Searching other sources

In addition to searching electronic databases, we identified process evaluations through examining included studies from a concurrent systematic review and meta-analysis of RCTs that explored the effects of interventions in reducing sedentary behaviour, using the same eligibility criteria for participants, interventions and comparators (Hall *et al*[13]). For each included study in the systematic review and meta-analysis of RCTs, we identified whether a process evaluation was conducted alongside the RCT and included all those identified. If the process evaluation results were not available, we contacted study authors for results.

### STUDY RECORDS
### Data management

References identified from electronic databases and other sources were deduplicated and imported into Endnote V.X7 reference management software. References were then imported in to Covidence (www.covidence.org, 28 April 2021), a web-based systematic review tool.

### Selection process

Using Covidence, two reviewers (RC and NL) independently assessed titles and abstracts of records from the electronic searches against the eligibility criteria and excluded obviously irrelevant studies. The full text of the remaining studies were obtained; then independently assessed, by the same reviewers, against the eligibility criteria to determine which studies would be eligible for inclusion. The same process for updated literature searches was undertaken (by NL and SO). During the screening process, disagreements were resolved by a consensus-based decision between the reviewers, or if necessary, discussion with a third reviewer (DC).

### Data extraction and narrative synthesis

A narrative approach to synthesising data was undertaken to provide detailed written commentary to address the research aims and objectives. Reviewers (RC, NL and JFJ) independently extracted relevant quantitative and qualitative data from included studies. All quantitative data was checked by a second reviewer (SO). Fifty per cent of the qualitative data was compared by NL and JFJ.

### Developing and refining the framework

To direct data extraction, a framework was produced based on this review's aims, objectives and data to be extracted as specified in the protocol.[11] The six themes and relevant subthemes align with the key functions in the MRC process evaluation framework[4] (table 1). Data extraction items (related to the trial and process evaluations)[11] were coded into the framework then summarised in a series of files focusing on: the characteristics of trials (online supplemental file 3), characteristics of process evaluations (online supplemental file 4), delivery methods and mechanisms of impact (online supplemental file 5) and implementation data including fidelity, recruitment, retention and reach (online supplemental file 6). Within online supplemental file 6, we have included definitions of these terms; informed by three key papers.[4 14 15] Qualitative data from the framework is presented in the 'narrative synthesis findings' section.

To help understand the effects of each included intervention on sedentary behaviour outcomes, the sedentary behaviour measures from the associated RCTs were also extracted (online supplemental file 7). As the review focuses on the findings from the process evaluations, the treatment effects estimated in the RCTs have not been synthesised or analysed.

Two reviewers (JFJ and NL) independently coded one study to pilot the framework. Following discussion, minor refinements were made before the final framework was agreed. For example, engagement was added in to barriers and facilitators to participation in the intervention, a clearer definition of context was added and a sixth 'miscellaneous' theme was included to code data about trial procedures and qualitative methods, mainly for context where appropriate. The coding rules were also refined, then used in coding the remainder of the included studies.

### Coding into the framework

Using the framework, JFJ independently coded all included studies. Nine studies (every other study listed alphabetically) were coded independently by NL. Coding was managed using NVivo software V.12 Plus.[16]

### Comparing codes

JFJ and NL compared data from the nine studies coded by both researchers. To enhance the rigour of the process, JFJ then re-reviewed all studies coded singly to ensure consistency.[17]

### Methodological quality

Methodological quality of included studies was assessed using the Mixed Methods Appraisal Tool (MMAT),[18] which is designed to concurrently assess qualitative, quantitative, and mixed methods studies. Three reviewers (NL, RC and JFJ) independently assessed the quality of studies and resolved any discrepancies by making a consensus-based decision, or if necessary, by discussion with a fourth reviewer (DC). Studies were not excluded from the synthesis based on the outcome of the quality assessment.

**Table 1** Coding framework

| Themes and subthemes | Definition/descriptions of what should be coded |
|---|---|
| **1. Implementation data** | |
| 1a. Intended delivery | How the intervention was intended to be delivered (in main paper or protocol). |
| 1b. Actual delivery (including when adapted) | How the intervention was actually delivered, including when it has been adapted from what was intended. |
| 1c. Strategies for achieving delivery | How the intervention delivery was achieved (eg, tailoring interventions to individuals). |
| 1d. Measures of adherence | A measure of adherence that was used in the study (NB: may be some overlap with compliance/fidelity). Definition adopted: 'The extent to which delivered content, frequency, duration and coverage of intervention components/ material are as intended.' |
| **2. Mechanisms of impact** | |
| 2a. Logic models used to explain how the intervention was intended to work | Coded when a logic model is present. |
| 2b. Theories underpinning the intervention | Theories underpinning the intervention for example, transtheoretical model, social cognitive theory and behavioural change techniques (BCTs) from the 93-item taxonomy used as part of the intervention for example, goal setting, self-monitoring. NB: still coded BCTs even if authors do not make reference to a BCT taxonomy. |
| 2c. Mediators of change | Factors that explained how the intervention had an effect. |
| 2d. Responses to and interactions with the intervention | Instances where participants or those providing the intervention talked about how they responded to, or interacted with the intervention. |
| 2e. Intended mechanisms of action influencing intervention effectiveness | How the intended mechanisms of action influenced effectiveness (eg, intended mechanism of effect- self monitoring of daily activity). |
| 2f. Unintended mechanisms of action influencing intervention effectiveness | Descriptions of how unintended mechanisms of action influenced effectiveness (eg, if social support increased intervention effectiveness but the intended mechanism was self-monitoring). |
| **3. Contextual factors influencing effective and ineffective interventions (Context includes anything external to the intervention that may act as a barrier or facilitator to its implementation or its effects[4]).** | |
| 3a. Influencing implementation | Anything external to the intervention that may have influenced its implementation. |
| 3b. Influencing mechanisms | Anything external to the intervention that may have influenced the mechanisms by which the intervention had an effect (or not). |
| 3c. Influencing outcomes | Anything external to the intervention that may have influenced the outcomes of the intervention. |
| **4. Barriers and facilitators** | |
| 4a. Barriers to delivery of intervention | Factors that hindered the delivery of the intervention (including internal factors). |
| 4b. Facilitators to delivery of intervention | Factors that enhanced the delivery of the intervention (including internal factors). |
| 4c. Barriers to participation and/or engagement in intervention | Factors that hindered participation or engagement in the intervention: 'The extent to which participants understand, accept and enact specific components of the programme in their daily lives.' |
| 4d. Facilitators to participation and/or engagement in intervention (eg, incentives) | Factors that enhanced the delivery of the intervention. Definition as above. |
| 4e. Recommendations made to address barriers and facilitators. | Recommendations made to overcome the barriers and facilitators (from either the study participants (including those delivering)) or the authors of the paper. |
| **5. Understanding and experiences of interventions from different perspectives.** | |
| 5a. Participants' experiences | Experiences from the perspectives of participants that cannot otherwise be coded into context, or barriers and facilitators (likely to be direct quotations). |
| 5b. Family and carers' experiences | Experiences from the perspectives of family and carers that cannot otherwise be coded into context, or barriers and facilitators. Carers defined as unpaid and informal carers so includes friends and relatives but not paid carers. |
| 5c. Staffs' experiences | Experiences from the perspectives of staff that cannot otherwise be coded into context, or barriers and facilitators. Paid carers that are involved in the intervention would be included here. |
| 5d. Control group experiences | Experiences from control group participants if reported. |
| **6. Miscellaneous** | |
| 6a. Trial procedures data | Instances where study includes information that is more focused on the data collection for example, recruitment and retention, rather than the intervention. Agreed not to code any quantitative data that is otherwise captured elsewhere in the review. |
| 6b. Qualitative methods (to provide context) | Reports of how qualitative data collection was undertaken for example, 'semistructured interviews were conducted with 10 staff.' |

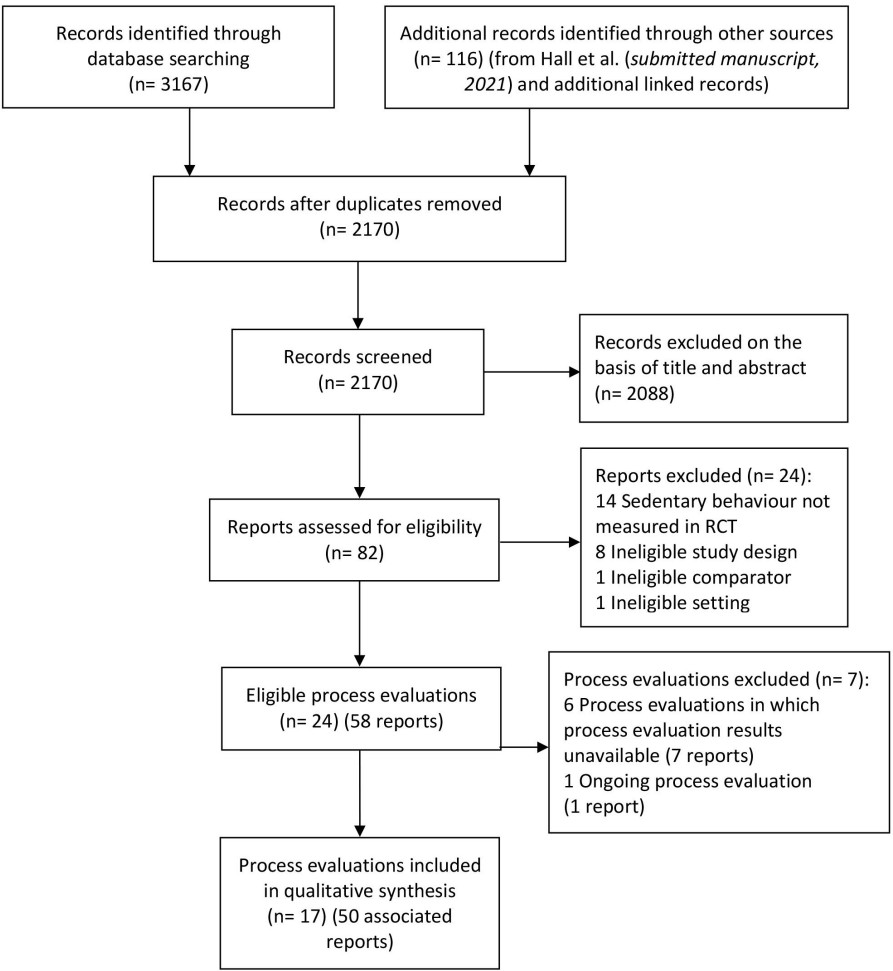

**Figure 1** PRISMA flow diagram. PRISMA, Preferred Reporting Items for Systematic Reviews and Meta-analyses; RCT, randomised controlled trial.

## RESULTS

The PRISMA flow diagram (figure 1) presents results from all searches. Database searches identified 3167 records; 116 additional records were identified through other sources. After removing duplicates (n=1113), 2170 titles and abstracts were screened; 2088 records were excluded as they did not meet the predefined eligibility criteria. The full-text reports of the remaining 82 records were assessed for eligibility, of which 24 reports were assessed as ineligible. The results of process evaluations of six eligible studies (seven reports) were unavailable. In total, 17 process evaluation reports were included for data synthesis. Fifty associated reports were also retained to address objective one.

### Record of excluded studies

Online supplemental file 8 provides reasons for excluding the 24 studies outlined in figure 1.

### Summary of included studies

#### Included RCTs

To address objective 1, and provide context for the process evaluations, online supplemental file 3 presents data from trials with included process evaluations, including: aims, inclusion/exclusion criteria, sample size, participant characteristics, study design, intervention and control descriptions, data collection and follow-up time points and outcome measures used.

### RCT aims

Associated trials where sedentary behaviour was measured as an outcome were published between 2007 and 2020. Five trials focused specifically on reducing sedentary behaviour[19–21] or sitting time.[22 23] The remaining 12 trials aimed to increase physical activity or promote healthier lifestyles but measured sedentary behaviour as an outcome (online supplemental file 3).

### Trial location and participant characteristics

Seven trials were conducted in the UK,[20–22 24–27] the remainder in the USA,[19 23 28] Netherlands,[29 30] Brazil,[31] Ireland,[32] Canada,[33] Hong Kong[34] and Belgium.[35] Participants recruited into the trials varied, including: mothers or parents of infants, pregnant women, adults, older adults, overweight adults, individuals with chronic illnesses and individuals with intellectual disabilities or serious mental illnesses. Most trials included males and females, however, three included females only.[19 26 28]

Participants' ages ranged between 30 and 75; the majority of trials included participants aged between 40 and 50 years.[19–21 24 25 29 30 32 33] Only nine trials reported ethnicity, the most ethnically diverse study was by Albright *et al*[28] which reported the following ethnicities: Native Hawaiian, Pacific Islander, Asian, mixed race, white, black-Native American.

### Included process evaluations

Online supplemental file 4 presents data specific to the process evaluations including: aims and whether process evaluations were prespecified, sample size and sampling methods, study design and data collection methods, and theoretical frameworks used. These data provide further context for the narrative synthesis.

Thirteen process evaluations were pre-specified in published protocols, or trial register records. Five studies[19 26 30 32 35] were published prior to the MRC guidance,[4] which was developed to provide a more systematic approach for planning and conducting process evaluations. The majority were published in the same year or after the guidance was published.[20–25 27–29 31 33 34] Despite this, nine studies did not report using a framework or guidance[19 21 23 24 28 32–35] only four authors cited the MRC guidance[20 22 25 27] and only one reported using this to guide the process evaluation.[25] As shown in online supplemental file 4, five studies cited other frameworks[20 26 29–31] the most common alternative to the MRC framework being the RE-AIM (Reach, Effectiveness, Adoption, Implementation, and Maintenance) framework.[36] Fourteen used the term 'process evaluation' within the publication. Three did not use this term.[23 24 34]

### Process evaluation aims

There was considerable variation in process evaluation aims. Some studies had a broad focus on participants' experiences, for example, Elramli[24] aimed to explore participants' views regarding the effectiveness of a walking intervention for rheumatoid arthritis (RA). Others focused more specifically on barriers to achieving activity goals,[28] or barriers and facilitators to the sustainability of an intervention.[29] Some focused on the feasibility and/or acceptability of interventions among different participant groups, including those at risk of chronic disease;[33] older adults;[23] individuals with intellectual disabilities;[20] and individuals with serious mental illnesses.[21] Only two process evaluations were conducted with a view to refine the intervention.[26 27]

### Study design and data collection methods

As outlined in online supplemental file 4, sample sizes of participants recruited to the process evaluations varied, from 5[21] to 411.[29] A total of 1553 participants were included from intervention groups across the 17 studies and 340 from control groups in four studies.[26 29 34 35]

Nine studies[19–22 25–27 29 31] used mixed-methods, most commonly combining quantitative questionnaires with semistructured interviews (telephone and face to face).

In five studies, questionnaires were used to ask participants about their satisfaction with the intervention, intervention fidelity, and about suggested improvements to interventions.[19 27 29 32 35] In two studies, questionnaires focused on intervention providers' experiences of delivering and participating in interventions.[25 30]

Semi-structured interviews explored intervention contexts, barriers and facilitators to intervention delivery, and experiences from the perspective of intervention providers, participants, and their family members or carers.[21–27 29 31 33 34] Other methods used included: non-participant observations,[19] focus groups,[20 25 27 31 34] healthcare professionals' registries and log books.[29]

### Methodological quality

Online supplemental file 9 provides an overview of answers to questions in relevant categories of the MMAT[18] for all included studies. Options include, 'yes', indicating a positive judgement, 'no', indicating a negative judgement or 'can't tell,' which is used when there is insufficient information to make a judgement. MMAT authors discourage calculating an overall score and excluding studies based on their methodological quality.[18] Therefore, all studies remained included in the synthesis and were not weighted. Below is a summary of the assessment of each of the six categories.

### Screening questions

The majority of studies had clear research questions or aims, and appropriate data were collected.

1. Qualitative studies

   Thirteen of 17 included studies had a qualitative component. Four[21 26 27 34] were rated as not meeting some of the criteria in this category, because descriptions of the analysis process lacked detail, and it was unclear how authors arrived at their findings. In these studies, findings were commonly presented as a series of quotes, in tables or online supplemental files but interpretation was considered too limited to constitute an in-depth analysis.

2. Randomised controlled trials

   Each of the included studies was associated with an RCT. This category of the MMAT was used to assess the quality of the trials. The 'can't' tell' option was most commonly used in this section because authors often provided insufficient information to provide an answer, particularly regarding the randomisation process and blinding. Scoring was more mixed within this category and no studies scored yes for all questions.

3. Non-randomised studies

   The associated trials were all RCTs; therefore, this category was not applicable.

4. Quantitative descriptive studies

   Thirteen studies had a quantitative component. Overall, they were rated positively across all questions.

5. Mixed-methods studies

   We considered studies which used methods meeting the criteria for both categories 1 and 4 as mixed-methods

Factors influencing context that facilitate or hinder implementation or how participants respond or interact with the intervention (Review objectives 5 & 6)
- **Barriers to delivery of interventions:** administrative or scheduling issues, organisational challenges, absence of staff , closure of services, work pressures for staff, financial **difficulties**
- **Barriers to participation and engagement:** most common examples included: pre-existing illness or injury e.g. pain, other commitments e.g. work or caring responsibilities, being too tired. Other examples included difficulties using accelerometers or pedometers, technology problems, context specific expectations or norms, being worried about specific environments e.g. gyms due to safety risks, influence of usual everyday life routines, religious festivals at certain times of year
- **Facilitators to the delivery of interventions:** simplicity in structure of the programme, little need for preparation in addition to usual working requirements, having access to trial related resources, having a committed team
- **Facilitators to participation and engagement:** most common example- support and encouragement from providers and peers to keep on track – mixed preference for one to one vs face to face interactions. Other examples- being accountable to someone, having accurate step monitors, access to textual resources
- **Experiences of interventions from different perspectives:** positive experiences included opportunities for learning (participants and staff), becoming more aware of sedentary behaviour (participants and staff), negative experiences included disliking particular part of intervention (participants), or limited space for delivering programmes (staff)

KEY FINDING – A range of contextual and other factors influenced whether the interventions could be implemented or how participants responded. These need to be considered so that interventions can be adapted to different contexts and participant groups

**Description of the interventions and their causal assumptions** (review objectives 1 and 2)
- All interventions have multiple components (See supplementary file 5)
- Group based input or support common
- Delivered by a range of providers e.g. researchers, health educators, exercise professionals, coaches, advisors and nurses
- All underpinned by theory or incorporated BCTS- most common theory – Social Cognitive Theory (SCT)

KEY FINDING: Only one study included a logic model outlining causal assumptions

**Implementation and delivery approaches** (review objective 3)
- Interventions delivered as intended in 3 studies
- 7 interventions were adapted, 7 difficult to determine
- Approaches for achieving intervention delivery:
  - Staff training
  - Tailoring interventions to individual needs
  - Allowing for flexibility in delivery methods

KEY FINDING- enhancing fidelity with adaptations did not always lead to interventions being effective in achieving the intended outcomes. Need to understand more about the mechanisms of intended effect

**Mechanisms of impact influencing intervention effectiveness** (Review objectives 2 and 4)

- Mechanisms of action were reported, examples related to the most commonly used theory (SCT) included: self-efficacy, behavioural cues, using resources e.g. websites and counselling calls, social support
- Variation in how much these intended mechanisms had an effect on reducing sedentary behaviour. For example in one study, social support did not have the intended effect for those with intellectual disabilities as it was not appropriate for this population

KEY FINDING – it is important to fully understand the context and complexities associated with how interventions will work to understand more about the mechanisms of effect

**Outcomes**
Reducing sedentary behaviour

**Figure 2** Key findings mapped to the diagram from the Medical Research Council (MRC) guidance for process evaluations.

studies. This category was only applicable for nine studies. When studies were rated negatively on either the qualitative or quantitative component, it was reflected in the judgement for this category.

## Narrative synthesis findings
This section reports on the findings from the 17 process evaluations coded into the framework and summarised in narrative form. Subheadings based on the key functions of a process evaluation outlined in MRC guidance by Moore *et al*[4] have been applied to organise the data. Figure 2 (based on Moore *et al*[4]) outlines summary findings for each subheading in the synthesis and identifies some key findings.

### Description of the interventions and their causal assumptions
According to Moore *et al*,[4] a clear description of the intervention and its causal assumptions are an important part of understanding how other factors (eg, implementation, context and mechanisms of impact) influence outcomes.

Online supplemental file 5 describes the content and delivery methods for all interventions. Intervention delivery periods ranged between 6 weeks and 18 months. All interventions included multiple components, examples include group based educational sessions combined with email input and self-monitoring tools[19] or one-to-one counselling combined with tailored email input.[28] In terms of delivery, interventions commonly incorporated some group based input or support.[19 21 22 24–26 29 31 34] Interventions were delivered by a range of providers including researchers,[19] health educators,[22 28] exercise professionals, including personal trainers,[20 29] coaches,[21 23 33] advisors and nurses.[25 30]

Online supplemental file 5 also includes information about the mechanisms by which the interventions are intended to have an effect, and any theoretical underpinnings. All interventions were underpinned by theory or incorporated behavioural change techniques, the most common theory being social cognitive theory.[37]

### Implementation and delivery approaches
Moore *et al*[4] recognise that interventions can have limited effects due to weaknesses in how they are designed, or because they are not properly implemented. This section outlines the extent to which interventions were reported to be delivered as intended, common approaches used in intervention delivery, and whether this reportedly translated into changes in outcomes.

As indicated in online supplemental file 5, in three studies[21–23] interventions were reportedly delivered as intended. In seven studies,[19 20 25 28–30 33] adaptations were made to the interventions during the course of the trial. In the remaining seven studies,[24 26 27 31 32 34 35] it was difficult to determine whether there were any adaptations as authors only reported the actual delivery, not the intended delivery.

Approaches for achieving intervention delivery included: ensuring staff were appropriately trained and prepared to deliver the intervention with fidelity;[19 31] tailoring aspects of the programme to individuals and their needs (eg, ensuring activity consultations are appropriate for those with intellectual disabilities[20]); and allowing for flexibility in delivery methods. For example, in Poston et al[26] pregnant women were provided with the option of receiving the intervention via phone or email, rather than sessions delivered at the hospital, and in Berendsen et al[29] coaching meetings as part of the intervention were planned with consideration of holidays and health issues.

Despite these adaptations for enhancing fidelity, interventions were not always effective in achieving the intended outcomes. For example, in Poston et al[26] despite flexibility in the delivery mode, objectively measured physical activity and sedentary behaviour did not change in the intervention group. In this particular participant group (pregnant women), the potential to achieve the targeted health outcome, optimal blood glucose level, via dietary changes, was greater than changes in physical activity, including sedentary behaviour, as for some participants increasing their activity led to feelings of discomfort. Similarly, in Matthews et al[20] although individual tailoring was used, the intervention did not have a significant effect on any of the primary or secondary outcomes including time spent in MVPA and time spent sedentary. It was suggested that this intervention may need to be longer than 12 weeks for individuals with intellectual disabilities. This highlights the importance of understanding more about how an intervention is intended to have an effect, as outlined in the following section.

### Mechanisms of impact influencing intervention effectiveness

Moore et al[4] emphasised the importance of exploring mechanisms through which interventions bring about change, to learn more about how the intervention effects may have occurred and how they may be replicated in similar future interventions. This section outlines the mechanisms reported across the studies and the extent to which they impacted on behaviour and outcomes.

Social Cognitive Theory was the most commonly used theory, and the following mechanisms of action were reported in several studies: enhancing self-efficacy by rating confidence in completing goals;[19] using behavioural cues, for example, standing up every hour, and leaving the remote at the TV;[19] using resources, for example, websites combined with counselling calls to encourage goal setting[28] providing social support in educational sessions or workshops, and input and engagement from carers.[19 20 22 24 28]

However, across the studies, the extent to which these mechanisms had their intended impact on behavioural change varied. In Elramli[24] the intervention aim was increasing daily step count, social support was found to be a key factor in participants who increased their physical activity. However, behavioural change techniques including social support, feedback and self-monitoring were to a lesser extent associated with reduced sedentary behaviour in those with RA. In Matthews et al[20] where the intervention aimed to increase walking and reduce sedentary behaviour, the social support component was not effective for adults with intellectual disabilities. In Biddle et al[22] where the intervention aimed to reduce sitting time, there was no difference in sedentary time at 12 months between intervention and control arms. Reasons for a lack of change in sedentary behaviour included: a preference for adopting physically active behaviours rather than sitting less, and motivational drift after 3 months. In Adams and Gill[19] which focused on reducing sedentary behaviour and increasing light physical activity, self-efficacy was not shown to be a predictor of change in sedentary behaviour. Behavioural cues, for example, leaving the remote at the TV, did not always influence behaviours either, because some participants were already doing the cued behaviour, and some did not have a TV.[19]

Studies underpinned by the Transtheoretical Model, Theory of Planned Behaviour and Self-Determination theory placed emphasis on encouraging participants to be aware of and monitor their own behaviour.[20 29 30] Motivational interviewing was used in two studies to prompt participants to find solutions, rather than telling them how to change their behaviour.[29 30] Berendsen et al[29] found the feasibility of changing physical activity behaviours and dietary habits was not as high as expected and was likely associated with poor adherence. Some participants were unrealistic about how much of their own effort would be required, which influenced attendance at meetings. Lakerveld et al[30] reported that practice nurses were competent and confident in the delivery of motivational interviewing and participants' satisfaction was high, but even so, almost no effects were seen in the determinants of behavioural change in this population of individuals who were at risk of cardiovascular disease and diabetes.

In summary, these findings provide some insights into how mechanisms may or may not have an effect on sedentary behaviour, highlighting that it is important to fully understand the complexities of interventions.

### Factors including context that facilitate or hinder implementation or how participants respond or interact with the intervention

Moore et al[4] regard understanding context as an important part of interpreting factors influencing whether interventions are effective. They defined context as anything external to the intervention that may act as a barrier to its implementation or effects. They also considered participants' responses to and interactions with the intervention as important mechanisms that could influence outcomes. Drawing on the coding framework, this section is divided into include barriers and facilitators to delivery of interventions, barriers and facilitators to participation and engagement, and understanding of participants experiences from different perspectives.

## Barriers to delivery of interventions

Across the studies, there were a range of barriers to delivering interventions, including administrative or scheduling issues and organisational difficulties or challenges. In two studies, planning educational sessions around other commitments including holidays and childcare responsibilities was difficult for staff.[24 34] In Blunt et al,[33] a central research team were involved in scheduling appointments, intending to reduce the workload for coaches. However, this resulted in increasing time spent scheduling and it was recommended that coaches were best placed to take responsibility for their own scheduling.[33]

Organisational difficulties were apparent across two studies.[20 31] A community health worker from one of the six health centres in Benedetti et al[31] described the long absence of a doctor as a turbulent time in the unit, which added difficulties in trying to deliver the intervention. In Matthews et al[20] the intervention was implemented at a time of significant change within the local learning disability service. Provision of support was affected by the closure of many day centres, which led to a low morale and increasing work pressures among the staff. In Berendsen et al[29] there were factors that influenced adherence; additionally suspended government financial and policy support meant the programme could not continue.

## Barriers to participation and engagement

Across the studies, there was a range of barriers to participation and engagement in the interventions. The most common barriers to engagement were: having a pre-existing illness or injury and associated problems, for example, pain,[19 23–29 33] having other commitments, for example, work, caring responsibilities;[23 24 26 28] and being too tired.[22 26 33] Other, less common barriers to engagement included loss of accountability for behaviour over time,[33] fluctuating mental health[21] and lack of motivation.[24]

Some participants also experienced difficulties with pedometers and accelerometers used as an outcome measure for the trial, in terms of understanding how to use them, side effects of wearing them, for example, skin irritation[19 23] and lost devices.[19 22] In Biddle et al,[22] half the participants experienced problems with the software for the 'Gruve' accelerometer, including: computer synchronisation issues, incompatible computers, website navigation problems, device malfunction, short battery life and charging issues.

Some barriers may be more applicable to specific groups. For example, in Benedetti et al[31] a community health worker perceived some older people to be apprehensive about new things which may have been a barrier to participation. In another study, a participant thought that sitting was deserved in old age and he was looking forward to this aspect of retirement to indulge in some of his passions, for example, reading and studying, which made him resent the idea of standing more.[23]

Some barriers were specific to particular contexts. In Elramli,[24] participants who had RA worried about using the gym because they lacked knowledge of suitable, safe exercises. Although workplace interventions were not included in this review, participants who had received educational based interventions reflected on how this applied to other parts of their lives, and therefore, provided some insight into how the work setting impacts on sedentariness. For example, participants felt that it was not appropriate to be standing in a work context which could cause embarrassment, for example, the expectation to be seated for meetings.[19 22 23] Further barriers at work included having no access to stairs and no standing desks.[22]

The context of other parts of everyday life was also influential for some participants who had developed ingrained sedentary habits, as a result of their usual activities or hobbies, for example, reading, eating, socialising, TV viewing and knitting.[23] Religious festivals had an impact on willingness to reduce sitting time at certain times of the year, for example, Christmas and Ramadan.[25]

## Facilitators to the delivery of interventions

Some of the approaches for achieving implementation and delivery could be regarded as facilitators, including: allowing flexibility in delivery methods, tailoring aspects of the programme to individuals, initial preparation and planning. A range of other factors facilitated intervention delivery.

For example, in Blunt et al[33] coaches valued the simplicity and structure of the programme. They also appreciated that the programme did not require extensive background knowledge or preparation over and above their existing working requirements. Coaches had the option of referring back to the Canadian Physical Activity Guidelines to ensure they were providing the right level of support to participants. In another study, not requiring too much additional trial focused expertise, and having access to useful trial related resources was valued by social workers.[34] In this study, the research team prepared and organised most of the materials which facilitated delivery. As a contrast to low morale among staff,[20] having a committed team was also important for facilitating delivery.[34]

## Facilitators to participation or engagement in intervention

There were a range of facilitators to participation and engagement in the interventions. The most common facilitator was support and encouragement from providers and peers; participants valued personal interaction and having someone to keep them on track with the intervention.[20 24 25 27 31 33]

In some studies, group environments facilitated engagement and provided opportunities for sharing experiences and meeting other peers in a similar situation.[21 24 27] In Matthews et al[20] many participants liked one-to-one engagement with intervention providers. This was particularly beneficial to the group who had intellectual disabilities, partly because the conflicting needs of participants in group activities were occasionally disruptive. This

group faced challenges to engagement with the intervention, compared with the general population. Matthews *et al* suggested the need for providing interventions to people with intellectual disabilities for longer than 12 weeks, so that consultations with providers can address more barriers.[20]

Being accountable to someone, for example, a health coach, also facilitated engagement in three studies because the participants felt being monitored provided motivation.[20 23 25 33] While use of a step count monitor was a barrier for some, others found this was a good motivator.[23 24] Adams and Gill[19] recommended that in order for pedometers to be beneficial they need to be more accurate. It was also suggested that technology should be tailored to detect movement in older adults which may be different from younger adults.[23]

Participants valued textual resources that were considered attractive through using appropriate text and images.[20 31] Adams and Gill[19] made recommendations for making resources more accessible including embedding videos in emails rather than asking participants to use YouTube, and printing cue cards out rather than asking participants to do so themselves. Less common facilitators were: already being involved in health programmes,[33] and becoming more aware of the extent of their own sedentary behaviour.[23]

### Understanding experiences of interventions from different perspectives
#### *Participants*
There was some overlap in data coded into barriers and facilitators and participant experiences. The experiences can be divided into positive and negative. Examples of common positive experiences included enjoyment or satisfaction with the intervention programme.[19 21 31] In some studies, participants described this as life-changing[23 25] or a new opportunity for learning about how to reduce sedentary behaviour and exercise safely.[24] As a result of engaging in the intervention, some participants recognised they had become more aware of the importance of reducing sedentary behaviour[19 24 31] and associated benefits, for example, weight loss,[21 23] and reduced stress,[23 34] less fatigue,[23] less pain[24] and lower blood sugar.[19]

Examples of negative experiences included: feeling stressed or nervous due to wearing a pedometer and a need to check it frequently;[24] disliking a type of counselling session because they expected to follow suggestions;[30] and feeling nagged by carers to participate.[20]

#### *Family/carers*
Only two studies included data regarding the experiences of families or carers.[20 34] There was a distinction between the carers' or family members' perceptions of participants' experience and their own experiences as part of an intervention or supporting the intervention. In Matthews *et al*,[20] family carers talked about how much the participants enjoyed their experiences due to reaching their goals and getting a certificate.

The dynamic was different in another study which included a family-based exercise intervention.[34] Participants valued reminding each other as a family to do their exercises.

#### *Staff*
There was also some overlap in data coded into barriers and facilitators and staff experiences. Most staffs' perceptions of participants' experiences were positive. In two studies, staff perceived participants enjoyed using pedometers and diaries.[20 25] Staff voiced positive perceptions of the programme, for example, encouraging others and themselves to fit physical activities into their everyday lives,[33] and enhancing the participants' family cohesiveness.[34] Being involved in delivering the programme also had benefits for some staff. It helped them understand the complexities associated with having a healthy lifestyle;[33] and reminded them to stand and move more in their own roles.[34]

Some negative experiences overlapped with the barriers to delivering the interventions. These included difficulties with staffing when they were already overcommitted;[20 31] limited venue space for delivering the programme;[31] and lack of psychological training to be able to deliver the intervention.[29]

## DISCUSSION
### Summary of findings
This review aimed to synthesise process evaluations of interventions in trials where sedentary behaviour was measured as an outcome to: develop an understanding of intervention content, mechanisms of impact, implementation and delivery approaches and contexts, in which interventions were reported to be effective or ineffective and explore the experiences of participants, family/carers and intervention staff in such interventions. To address these aims, we synthesised data from 17 studies including a range of participant groups for example, mothers or parents of infants, pregnant women, adults, older adults, overweight adults, individuals with chronic illnesses including RA, intellectual disabilities and serious mental illnesses.

Systematic reviews of process evaluations have been conducted in other areas of research, for example, primary care[38] and workplace health promotion programmes.[39] However, to our knowledge, this review is the first to synthesise data from process evaluations of interventions in trials which measured sedentary behaviour as an outcome in adults.

The review has highlighted the complexity of factors that contribute to implementing interventions with fidelity, and how this links to outcome effects. Common barriers to delivery were those that may be expected in delivery of complex interventions of any kind, not just reducing sedentary behaviour. These included structural changes and staffing pressures within an organisation, and limited funding for providing interventions. Many

interventions required some level of input from providers (eg, researchers, health educators, exercise professionals, coaches and health professionals) to deliver the programme, for example, scheduled exercise or education sessions. On the other hand, this limited flexibility of a structured intervention posed difficulties among some participants who had busy schedules and other priorities. In such cases, delivery was facilitated by providing different options for how the intervention is delivered, for example, via phone or email. However, flexible intervention delivery did not guarantee adherence to the intervention, because participants faced other barriers for example, discomfort during pregnancy, cognitive difficulties; these factors ultimately impacted on sedentariness.

While it was not our primary intention to synthesise the quantitative findings from the RCTs; the quantitative findings (summarised in online supplemental file 7), indicate only three studies reported a statistically significant reduction in sedentary behaviour at the end of the intervention.[21 24 33] The review identified commonalities across these three interventions that were effective in reducing sedentary behaviour; they all included elements of goal setting and access to support or coaching from a professional. All three were underpinned by theories (social cognitive theory of self-regulation, social cognitive theory and the COM-B model, including a focus on self-efficacy) which in part explain how these interventions may have had their effects (online supplemental file 5). However, other studies also had similar features, were underpinned by similar social cognitive principles including self-efficacy[19 22 26 28] but reported no statistically significant reduction in sedentary behaviour. In three of these four studies, control group participants still commonly received some form of information, for example, a leaflet or workbook which could be regarded as informational support. This may account for not finding a statistically significant effect when compared with the interventions, if their mechanisms of effect are quite similar. These findings identify that the process of changing outcomes, for example, sedentary behaviour is complex and influenced by other factors, aside from intervention components.

Complex interventions were traditionally understood as those comprised of multiple components.[3] However, context is becoming increasingly recognised as a source of complexity with acknowledgement that interventions are not a discrete package of components, but also a process of changing what complex systems do, including the interactions between individuals (eg, providers and recipients).[40] Our findings support this notion because while all interventions were underpinned by psychological theories focused on individual-level change, for example, social cognitive theory,[37] trans-theoretical model,[41] theory of planned behaviour,[42] self-determination theory[43] and habit formation theory;[44] it was evident that a range of wider, contextual factors in addition to individual factors also influenced the implementation and delivery of the intervention as part of complex systems. However, within the included process evaluations, programme theories

(including logic models) depicting how the intervention would operate in a particular context were rarely reported. Only one process evaluation reported a logic model.[25] Given the complex nature of the delivery and engagement associated with complex interventions, it is important that influences on outcomes such as reduced sedentary behaviour are understood as individual-level behavioural change processes, and in context, taking into account the complexities of experiences.[45] Ensuring logic models are developed and reported would aid in understanding these complexities.

The identified barriers and facilitators to participation and engagement provide important insights into participants' experiences of interventions and explain what makes interventions more acceptable to some individuals compared with others. The review indicates that social support was important. Some participants valued elements of groups such as meeting others and sharing experiences among similar peers. Others, particularly those with intellectual disabilities, valued one-to one input from providers. Level of motivation was also influential in engagement. Some felt motivated due to being accountable to someone; while others felt motivated as a result of tracking activity using a pedometer. However, others disliked pedometers because they struggled to understand the device or experienced skin irritation while wearing them. Previous studies have found satisfaction being important for compliance and engagement with tracking devices, for example, pedometers.[46 47] Results of a national cross sectional survey conducted in Australia suggested that interventions should make sure the devices align with the preferences of the target groups.[48] Our review suggests that individuals with particular conditions could benefit from interventions that are tailored to their symptoms for example, pain, tiredness and illness.

Changes across the lifespan should also be considered so that interventions can take into account what is appropriate and acceptable for older adults. Our review findings indicate that older people may be more likely to think that sitting down is deserved, or associated with enjoyable hobbies, for example, reading. A recent review by Compernolle et al[49] focused on older adults perceptions of sedentary behaviour similarly found that sedentariness was motivated by finding enjoyment and comfort. Their experiences are also shaped by their capabilities, the social opportunities, and motivations in addition to societal expectations that often dictate that for older people sitting is their main mode of living.

Current lifestyles, regardless of age or other characteristics also influence the extent to which participants are likely to engage in behaviours that reduce sedentary behaviour. Our review evidence adds to, and supports findings from another review exploring qualitative experiences of participating in non-workplace interventions.[9] Sedentary behaviour is further complicated by seasons and events for example, celebrations such as Christmas or Ramadan which disrupt normal behaviour patterns,

and perhaps lead to less concern with healthy behaviours, even with interventions. A systematic review of factors that influence physical activity and sedentary behaviour in ethnic minority groups in Europe also identified cultural and religious factors as influential in the extent to which individuals were sedentary.[50] However, they highlighted that aside from the celebrations and events, some parts of religious activity, for example, walking to religious sites for prayers actually facilitated reduced sedentary behaviour and increased physical activity. It is possible that people from different ethnicities may also experience sedentary behaviour and physical activity differently, however, it is difficult to determine based on the data available in this review given that only 9 of the 17 studies reported ethnicity, and only three of those nine provided commentary on ethnicity. Albright et al[28] identified that non-white racial or ethnic groups were less likely to meet their goals compared with white participants. Poston et al[26] and Harris et al[25] both included commentary on ethnicity in the context of recruitment to the trials and process evaluations. In Poston et al[26] the process evaluation included women in urban hospitals in areas where socioeconomic deprivation was high, they also highlighted that obesity rates are higher among those with lower socioeconomic status, less qualifications and African and black Caribbean groups. The relatively low uptake (one-third approached for recruitment) was consistent with other studies with low uptake in healthcare. Harris et al[25] reported that participation was only 11% among adults and older adults in a socially and ethnically diverse population, with lower rates in more deprived Asian subgroups. This limited the ability to investigate differential effects in important subgroups. These authors have not drawn firm conclusions about how ethnicity and race may effect outcomes but Harris et al[25] highlighted that differential uptake of interventions that are found to be successful in trials could lead to increases in inequalities in physical activity levels so this needs to be monitored.

Looking across the barriers and facilitators identified in this review and the wider literature, a range of factors need to be considered, highlighting how difficult it is to develop interventions that are suitable for participants, even those with apparently similar characteristics. The Consolidated Framework for Implementation Research is an example of a taxonomy of constructs, organised into five domains (intervention, inner setting, outer setting, individual characteristics and process) that has been devised to understand what influences implementation that could be applied to further understand such complexities.[51] Interventions require some level of adaptation to the context and may need to be tailored to participants, including those share similar characteristics, for example, those with RA or intellectual disabilities. They also need to consider the dynamic between staff, participants and families as part of working towards a shared goal (eg, reducing sedentary behaviour).

Proctor et al[52] outlined a conceptual framework to understand interrelated outcomes in implementation research including: (1) implementation outcomes, for example, appropriateness, sustainability and costs; (2) service outcomes, for example, safety and timelines; and (3) client outcomes, for example, satisfaction. This is another example of a framework which incorporates outcomes that are not already included in the MRC framework for process evaluations. This framework could be applied as part of understanding the complex dynamics of implementing and tailoring interventions and would assist in highlighting some of the challenges associated with tailoring interventions, for example, material and staffing resource limitations, and what might be required for sustainability.

Based on the current findings, if we are to reach a point where reducing sedentary behaviour becomes habitual once interventions cease, participants will need simple strategies and support to take ownership of their own behaviour so they can sustain the lifestyle changes within the context of their lives and their preferences.

### Strengths and limitations

This is the first systematic review to synthesise data from process evaluations evaluating interventions in trials that measure sedentary behaviour as an outcome in adults. Robust methods were used throughout the conduct of the review. A comprehensive search strategy was developed with input from an information specialist; two reviewers independently screened search results and assessed the quality of included studies.

Although a large proportion of the trials on which the process evaluations were based were conducted in the UK, the inclusion of studies from other countries (eg, USA, Netherlands, Brazil and Hong Kong) mean these findings are relevant for researchers internationally. The inclusion of males and females enhances the applicability of the findings in terms of gender. However, with regard to age, the majority of studies included participants between 40 and 50 years; therefore, not all findings are applicable to other age groups. The inclusion of participants from various groups can be regarded as both a strength and limitation of this review. Findings may be of interest to experts in different research areas; however, it is difficult to draw firm conclusions for particular population groups, especially where sample sizes are small.

There was an overall lack of consistency in how process evaluations were reported, this was also the case in a review of process evaluations in primary care.[38] Fourteen out of 17 used the term 'process evaluation' within the publication. Three did not use this term,[23 24 34] although they met the criteria for inclusion in that they aimed to explore participants' views on the factors that influence intervention effectiveness,[24 34] including the feasibility and acceptability of the intervention.

The assessments using the MMAT also indicated some variation in the quality of the process evaluations. The four studies that were considered lowest quality had poorer qualitative components[21 26 27 34] that lacked detail and depth, and had limited interpretation. When studies

were rated negatively on the qualitative component, it was reflected in the judgement in the mixed-methods category in the MMAT. Only four studies[20 22 25 27] cited the MRC guidance for process evaluations[4] but this did not always equate to better quality. Only one study by Harris[25] used the framework to guide the evaluation, whereas the other three only made reference to it in the introduction. Harris *et al*[25] was one of the higher quality studies overall, suggesting that using a framework to guide the whole process evaluation can be beneficial. However, the quality of the other studies that included frameworks such as RE-AIM[36] and Steckler and Linnan's[53] process evaluation framework was variable.

Figure 2 indicates that the studies reported a lot of data about the factors including context that influence implementation and how participants respond or interact with the intervention. However, only one process evaluation included a logic model outlining how the intervention intended to have an effect.[25] This means the theoretical understandings are more limited, making learning from previous evaluations more difficult. The importance of programme theories and logic models have been emphasised in recent MRC guidance,[54] researchers should incorporate this in future evaluations of complex interventions.

More than 24 tools are available to assess the quality of systematic reviews; however, there remains no clear guidance for which tool to use for assessing the quality of process evaluations.[55] The MMAT[18] was a logical choice as it is appropriate for mixed methods studies and those using either qualitative or quantitative data. However, it has not been designed to require detailed commentary about judgements of quality. Therefore, a simplified account of quality is presented. Yet, it is difficult to compare studies without looking across all the domains because the authors do not recommend calculating an overall score.[18] It was also recommended that studies should not be excluded based on their quality,[18] accordingly all studies were included in the synthesis. In our view there is also a need to develop guidelines specific to systematically reviewing process evaluations of complex interventions.

The initial searches for this review were conducted in May 2019 and were repeated in May 2020. We acknowledge that this area of research is experiencing considerable growth in numbers of publications. Studies published since May 2020 were not included in the current synthesis. Recognising this limitation, we repeated the searches in October 2021 using the same parameters as previously. We have presented these new searches in online supplemental file 10.

Overall 464 unique articles were identified once 14 duplicates were removed. Two reviewers completed title and abstract screening identifiying 29 for full-text screening; of these, 21 met our criteria, 8 are ongoing studies,[56–63] 8 are completed trials where a process evaluation was conducted but results are not available,[64–71] and 5 are completed studies with process evaluation results available.[72–76] As with the studies that were synthesised

in our review, these included participants from a range of different ages and health conditions for example, insomnia disorder, diabetes, heart disease, hip fracture and obesity and generally focused on increasing physical activity, reducing sedentary behaviour or were lifestyle or weight loss interventions.

Of the five eligible studies where process evaluation results are available, one study[72] was guided by the MRC framework,[4] none of the other studies used this or other frameworks to guide their evaluation. This study by Blackburn *et al* was the only one where the intervention (SITLESS) aimed to reduce sedentary behaviour in addition to increasing physical activity and physical function. The other four included a measure of sedentary behaviour but the intervention primarily aimed to increase physical activity[75 76] or promote lifestyle changes including weight loss.[73 74] These process evaluations have different aims, one explored older adults experiences of an intervention (SITLESS) which combined an exercise referral scheme plus self-management strategies,[72] one explored factors that support older people to increase their physical activity levels in a primary care based intervention (PACE-Lift);[76] one explored how participants of different ages with a range of conditions experienced and engaged with the e-coachER intervention which combined support and an exercise referral scheme;[75] one focused on the feasibility and satisfaction of an email lifestyle intervention aimed at minority breast cancer survivors;[74] another explored engagement and compliance with a community weight loss intervention for obese males (SHED-IT).[73]

The findings from our updated search demonstrate the growing literature on testing and evaluating interventions including understand the factors that influence experiences, engagement, compliance, satisfaction and how interventions are implemented. While the reported findings of these studies appear to be largely consistent with those included in our narrative synthesis, the iterative nature of coding data into the framework that was undertaken as part of this process means that it would not be appropriate to attempt to merge these findings into our already completed analysis. However, it is important to be aware of these recent studies when considering factors that influence how interventions focused on reducing sedentary behaviour are implemented, and how they are experienced.

## Conclusions

There is a wealth of existing evidence which synthesises the findings from trials evaluating interventions that have measured sedentary behaviour as an outcome in adults. This review complements existing trial evidence because it highlights a range of factors associated with implementation, context, and participants experiences that can impact on whether an intervention is effective or not.

It is promising that all interventions were underpinned by theory as part of understanding how they were intended to have an effect, however, it is important to acknowledge how different contexts and individual level

factors, for example, health status, illness, age and lifestyles can shape levels of engagement and behavioural change.

Researchers could benefit from using a process evaluation framework such as Moore *et al*'s,[4] for conducting and reporting process evaluations to ensure all factors are considered. Including logic model as part of the process evaluation would also assist in mapping the range of factors that contribute to changes in intervention outcomes.

**Author affiliations**
[1]Academic Unit for Ageing and Stroke Research, University of Leeds, Leeds Institute of Health Sciences, Leeds, UK
[2]Academic Unit for Ageing and Stroke Research, Bradford Institute for Health Research, Bradford, UK
[3]Faculty of Life Sciences and Health Studies, University of Bradford, Bradford, UK
[4]Geriatric Medicine, The University of Edinburgh Centre for Clinical Brain Sciences, Edinburgh, UK
[5]Faculty of Health and Medicine, The University of Newcastle School of Health Sciences, Callaghan, New South Wales, Australia
[6]Institute of Sport, Physical Education and Health Sciences, University of Edinburgh Physical Activity for Health Research Centre, Edinburgh, UK
[7]Institute of Psychological Sciences, University of Leeds, Leeds, UK
[8]Quality and Safety Research, Bradford Institute for Health Research, Bradford, UK

**Correction notice** This article has been corrected since it was published. In the methods section the initials RL has been replaced with RC (Rekesh Corepal referred in the acknowledgements).

**Acknowledgements** We acknowledge the help and support of our Information Scientist, Deirdre Andre, University of Leeds. We also thank Dr Rekesh Corepal (RC) for his contributions when the original search was conducted in 2019. We are grateful for the funding provided by the National Institute for Health Research (NIHR).

**Contributors** This systematic review was conceived and designed by members of the RECREATE Programme Management Group (AF, CE, CFF, RL and DC) and researchers (JFJ, NL, JH and SM). The systematic review process was conducted by JFJ, NL and SO with oversight and input from DC. JFJ drafted the initial manuscript with input from NL and DC. All authors have critically reviewed and revised different versions of the manuscript (JFJ, NL, SO, JH, SM, CE, CFF, RL, AF and DC). AF is acting as guarantor.

**Funding** This report is independent research funded by the National Institute for Health Research (Programme Grants for Applied Research, Development and evaluation of strategies to reduce sedentary behaviour in patients after stroke and improve outcomes, RP-PG-0615-20019).

**Competing interests** None declared.

**Patient consent for publication** Not applicable.

**Provenance and peer review** Not commissioned; externally peer reviewed.

**Data availability statement** All data relevant to the study are included in the article or uploaded as online supplemental information.

**ORCID iDs**
Jessica Faye Johansson http://orcid.org/0000-0003-3622-9598
Coralie English http://orcid.org/0000-0001-5910-7927
Rebecca Lawton http://orcid.org/0000-0002-5832-402X
David Clarke http://orcid.org/0000-0001-6279-1192

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
