## [Reviewer comments · BMJ Open]

ARTICLE DETAILS

TITLE (PROVISIONAL)	A SYSTEMATIC REVIEW OF PROCESS EVALUATIONS OF INTERVENTIONS IN TRIALS INVESTIGATING SEDENTARY BEHAVIOUR IN ADULTS
AUTHORS	Johansson, Jessica; Lam, Natalie; Ozer, Seline; Hall, Jennifer; Morton, Sarah; English, Coralie; Fitzsimons, Claire F.; Lawton, Rebecca; Forster, Anne; Clarke, David

VERSION 1 – REVIEW

REVIEWER	Craig, Kelly IBM Watson Health, Center for AI, Research, and Evaluation Employed by IBM Corporation
REVIEW RETURNED	28-Jun-2021

GENERAL COMMENTS	Overall reviewer comments: This article is well-written and has many strengths. The authors have met some expectations for the execution of a systematic review in this field; however, there are suggestions for improving and clarifying the methodology descriptions. There is value from the novel aspect of this work, identifying process evaluations of interventions in trials investigating sedentary behavior in adults. There are opportunities for the authors to improve the discussion and impacts of their research; generating additional visualizations, slicing the data to reveal demographic and/or geographic trends, and providing pragmatic next-best actions (with any recommendations specifically tailored for subgroups) to improve trial design and implementation are requested. Execution of methodology as strengths: 1. PROSPERO registration2. Strong adherence to PRISMA guidelines3. Detailed framework with working definitions to guide data extraction Recommendations: 1. METHODS - The search for this systematic review is out of date, as Supplementary file 2 indicates the search was executed through May 2020. It is best practice to update old systematic review searches, so they have reviewed the most recent literature (within six months). I recommend updating the searches.2. METHODS - Provide interrater reliability scores for full-text screening by the two independent reviewers as measured by kappa.
--

	3. METHODS - Provide search strategies in supplementary materials for each database used; only one was provided. 4. METHODS - Provide more information about your handsearching methodologies and the sources reviewed. 5. RESULTS - Supplementary File 3 – RCT study characteristics. Is there an easy way to differentiate your last column, outcome measures for treatment effects, based on what was pre-specified vs reported to indicate study intention? It would be helpful for others reviewing the summarized study content in the future to differentiate those studies with a primary goal of changing sedentary behavior within the table too (as you also present this info in the summary of included studies, p. 10); I feel it would support the reader, as it put context with the results outlined by the review authors. 6. FIGURES - If space available, it would be beneficial to have a visual figure illustrating the main findings. Within this figure it would be helpful to succinctly summarize your 6 review objective findings. Your study was successful and identified a load of data to process, and for the reader to synthesize themselves, as-is; if your findings could be conveyed as practical/pragmatic next steps for implementation of processes to decrease sedentary behavior, that would be especially meaningful. 7. SYNTHESIS/RESULTS - Given that the evidence had a limited study geography, it would be interesting to expand upon the racial and ethnic data that was captured as study participant characteristics. As public health initiatives are increasing (and are more broadly seen/heard in major media outlets), what considerations, if any, pertain to the examination of racial and/or ethnic or geographic (i.e., rural) disparities in sedentary behavior by these process evaluations identified? Can you expand on the effectiveness of these interventions in at-risk populations that have demonstrated and recognized health disparities (black, African-american, indigenous, Hispanic, rural vs urban, etc.); moreover, what worked/what didn't work and what were some of those barriers? Were some outcomes more effective in certain populations? Additional Comments: Line 9 on every page – for me anyways, the text font shifts
--	--

REVIEWER	Matthews, E Waterford Institute of Technology
REVIEW RETURNED	02-Jul-2021

GENERAL COMMENTS	Thank you for the opportunity to review this manuscript. This is important work and a fascinating read. Pg 3. 57-58 study design: This needs small expansion in my opinion. Notwithstanding that this review serves its intended aim at present, I have a point which pertains to the Aim of the research, but as a result has relevance to the entire review and is perhaps my salient and only point of query. I am wondering if 'to understand the intervention content', should be listed as the first aim within the research? To me, this seems a secondary point of relevance for this review, which is focused on process evaluations, or more specifically 'A SYSTEMATIC REVIEW OF PROCESS
--

	EVALUATIONS OF INTERVENTIONS IN TRIALS INVESTIGATING SEDENTARY BEHAVIOUR IN ADULTS'. One would hope that a review of sedentary behaviour intervention outcomes would, at least, also discuss intervention methods or issues such as barriers and facilitators as secondary outcomes. As an example, 'Intervention characteristics' are discussed in detail by another review of SB interventions by Gardner et al 2016. As a reviewer, one cannot help but feel that the intervention processes reported within studies are only as good as the quality of the process evaluations that have been carried out. Therefore, I think there is value in dealing with this in greater detail in this review, whilst not losing the other important aims. To give context to this point, the authors reference the work of Wierenga et al. 2013 as an example of a previous SR that has examined process evaluations. This work looks at primarily at the process evaluation in terms of the quality of these process evaluations and the findings with respect to 'implementation'. I think the article here submitted would benefit in shifting more focus on to the quality of process evaluations included given the title of the work. For further consideration, I am struck by the claims that seldom is qualitative data reported with rigour or specificity on data analysis. One also has to examine the supplementary file 4 for details of frameworks used in the included process evaluations, and little detail is extended to this point within the discussion. Again, given the title, and what may prompt one to read this article, prioritisation and expansion of this point, would in my opinion improve this article. Within strengths and limitations, where you make particular reference to age, I think it is also relevant to acknowledge the characteristic heterogeneity of populations such as those with intellectual disability and those with mental illness where clinical environments may exert strong influence on interventional engagement. The authors should mention in their discussion other implementation outcomes that may be relevant, but less explored through their framework but nonetheless, potentially relevant, such as outlined in the work of Proctor et al. 2011. Among included studies, Suppl file 3, Berendsen (2015): Can the authors clarify that the control conditions meet the inclusion criteria and do not extend to exclusionary. I didn't understand the nature of control conditions. There is perhaps a comment needed within the discussion about the 'intense' nature of some of the control conditions, in the context of limited efficacy of SB interventions. Overall Comments Some editing work required on spacing, punctuation and consistency of capital lettering throughout
--	--

VERSION 1 – AUTHOR RESPONSE

Reviewer: 1: Dr. Kelly Craig, IBM Watson Health Comments to the Author: BMJ Open-2021-053945 BMJ Open-2021-053945 Hall et. al., Interventions in trials investigating sedentary behaviour in adults	
Overall reviewer comments: This article is well-written and has many strengths. The authors have met some expectations for the execution of a systematic review in this field; however, there are suggestions for improving and clarifying the methodology descriptions. There is value from the novel aspect of this work, identifying process evaluations of interventions in trials investigating sedentary behavior in adults. There are opportunities for the authors to improve the discussion and impacts of their research; generating additional visualizations, slicing the data to reveal demographic and/or geographic trends, and providing pragmatic next-best actions (with any recommendations specifically tailored for subgroups) to improve trial design and implementation are requested. Execution of methodology as strengths: 1. PROSPERO registration 2. Strong adherence to PRISMA guidelines 3. Detailed framework with working definitions to guide data extraction	We would like to thank the reviewer for these positive comments. We acknowledge there are areas we need to revise to improve the paper and have outlined how we have addressed these below
Recommendations: 1. METHODS - The search for this systematic review is out of date, as Supplementary file 2 indicates the search was executed through May 2020. It is best practice to update old systematic review searches, so they have reviewed the most recent literature (within six months). I recommend updating the searches.	Thank you for this recommendation. Please see our earlier comment in response to the editors same suggestion.
2. METHODS - Provide interrater reliability scores for full-text screening by the two independent reviewers as measured by kappa.	Thank you for this comment. We have sought advice on this point and the Cochrane Handbook (version 5.1) for systematic reviews. Chapter 7.2.6 (Selecting studies – Measuring disagreement), states that kappa statistics are not recommended as standard in Cochrane reviews, because using arbitrary cut-points of kappa in comparison is unlikely to be useful in understanding the disagreements. We can confirm that the study selection was completed as described in the manuscript (page 4 lines 161-169); that is after the independent full-text assessment was completed in Covidence by two reviewers, they discussed the discrepancies in eligibility and reasons for exclusion to reach consensus according to the eligibility criteria. When uncertainty remained, they discussed with

	the third reviewer to reach agreement. This selection method is robust and effective for reaching consensus and based on the points made above we have not provided kappa scores.
3. METHODS - Provide search strategies in supplementary materials for each database used; only one was provided.	Thank you for this comment we have added all the searches to supplementary file 1 and also included another supplementary file 10 with our most recent searches.
4. METHODS - Provide more information about your hand searching methodologies and the sources reviewed.	Thank you for this comment, we have amended the searching other sources section on page 4 lines 162-169 to provide more clarity on our approach.
5. RESULTS - Supplementary File 3 – RCT study characteristics. Is there an easy way to differentiate your last column, outcome measures for treatment effects, based on what was pre-specified vs reported to indicate study intention? It would be helpful for others reviewing the summarized study content in the future to differentiate those studies with a primary goal of changing sedentary behavior within the table too (as you also present this info in the summary of included studies, p. 10); I feel it would support the reader, as it put context with the results outlined by the review authors.	Thank you for this comment, in Supplementary file 3, the last column has been renamed as “Outcome measures for treatment effects (identified in the study reports)”, to replace “Outcome measures for treatment effects (pre-specified or those only reported)”. The listed outcomes are those that are either included in the published study design report, and/or reported in the result reports. The study design or protocol is not available for all the included studies and we cannot confirm whether there were any changes from the original to the published study design to confirm which outcomes are truly pre-specified.
6. FIGURES - If space available, it would be beneficial to have a visual figure illustrating the main findings. Within this figure it would be helpful to succinctly summarize your 6 review objective findings. Your study was successful and identified a load of data to process, and for the reader to synthesize themselves, as-is; if your findings could be conveyed as practical/pragmatic next steps for implementation of processes to decrease sedentary behavior, that would be especially meaningful.	Thank you for this comment. We can see the value in having a visual representation of some of our key findings. We have produced a diagram (figure 2) based on one of the figures included in the MRC guidance for process evaluations (Moore et al., 2015) to highlight some of our key messages that relate to our objectives. We have added to the text in the narrative synthesis findings section on page 12 lines 384-385 where we have introduced the diagram. We have also made reference to figure 2 on page 20 line 779 of the discussion section to highlight the importance of programme theories, given these and logic models were lacking in the process evaluations included in our synthesis.
7. SYNTHESIS/RESULTS - Given that the evidence had a limited study geography, it would be interesting to expand upon the racial and ethnic data that was captured as study participant characteristics. As public health initiatives are increasing (and are more broadly seen/heard in major media outlets), what considerations, if any, pertain to the examination	Thank you for this comment, we have identified that there are nine trials where authors reported on ethnicity (which are documented in supplementary file 3). We have added some text to the ‘trial location and participant characteristics’ section on page 10 lines 301-303 which indicates the type of ethnicities that were included across the relevant trials. In the

of racial and/or ethnic or geographic (i.e., rural) disparities in sedentary behavior by these process evaluations identified? Can you expand on the effectiveness of these interventions in at-risk populations that have demonstrated and recognized health disparities (black, African-american, indigenous, Hispanic, rural vs urban, etc.); moreover, what worked/what didn't work and what were some of those barriers? Were some outcomes more effective in certain populations?	discussion, we have content on pages 18/19, lines 702-717 to address the point the reviewer makes about any considerations that may pertain to any racial or ethnic disparities in sedentary behaviour. Only three of the authors that reported data related to ethnicity or race included any commentary and this was mainly related to achieving goals or recruitment, therefore we do not have any data available which explicitly links ethnicity/race and sedentary behaviour but have included what is available. In relation to the second point made by the reviewer regarding the effectiveness of these interventions. As we indicated in our discussion section on line 640, page 17, whilst it was not our primary intention to synthesise the quantitative findings from the RCTs; the quantitative findings (summarised in supplementary file 7), indicate only three studies reported a statistically significant reduction in sedentary behaviour at the end of the intervention (Williams 2019, Elramli 2017, Blunt, 2018). None of these three studies reported ethnicity or provided commentary on any associations between particular characteristics accounting for their effectiveness. We did not aim to examine any other treatment effect outcomes in this review or our meta-analysis of RCTs (Hall et al., 2021), therefore we are unable to draw firm conclusions about whether some outcomes were more effective in certain populations. With regards to the point about what worked and what didn't, including barriers. We have provided a range of barriers and facilitators to delivery of interventions and participation or engagement in interventions throughout the narrative synthesis section. Where the data was available we have drawn out factors about particular groups e.g. older people possibly being more apprehensive about new things or thinking that sitting is deserved in older age.
Additional Comments: Line 9 on every page – for me anyways, the text font shifts	We have checked the revised manuscript for formatting issues and corrected them throughout.
Reviewer: 2 Dr. E Matthews, Waterford Institute of Technology	
Comments to the Author: Thank you for the opportunity to review this manuscript. This is important work and a fascinating read.	We would like to thank the reviewer for this comment; we are pleased the paper is a fascinating read.

Pg 3. 57-58 study design: This needs small expansion in my opinion.	Thank you. We have slightly expanded this section for clarity (page 4, lines 131-134) and it is now consistent with the content of the published protocol (http://dx.doi.org/10.1136/bmjopen-2019-031291).
Notwithstanding that this review serves its intended aim at present, I have a point which pertains to the Aim of the research, but as a result has relevance to the entire review and is perhaps my salient and only point of query. I am wondering if 'to understand the intervention content', should be listed as the first aim within the research? To me, this seems a secondary point of relevance for this review, which is focused on process evaluations, or more specifically 'A SYSTEMATIC REVIEW OF PROCESS EVALUATIONS OF INTERVENTIONS IN TRIALS INVESTIGATING SEDENTARY BEHAVIOUR IN ADULTS'. One would hope that a review of sedentary behaviour intervention outcomes would, at least, also discuss intervention methods or issues such as barriers and facilitators as secondary outcomes. As an example, 'Intervention characteristics' are discussed in detail by another review of SB interventions by Gardner et al 2016. As a reviewer, one cannot help but feel that the intervention processes reported within studies are only as good as the quality of the process evaluations that have been carried out. Therefore, I think there is value in dealing with this in greater detail in this review, whilst not losing the other important aims. To give context to this point, the authors reference the work of Wierenga et al. 2013 as an example of a previous SR that has examined process evaluations. This work looks at primarily at the process evaluation in terms of the quality of these process evaluations and the findings with respect to 'implementation'. I think the article here submitted would benefit in shifting more focus on to the quality of process evaluations included given the title of the work. For further consideration, I am struck by the claims that seldom is qualitative data reported with rigour or specificity on data analysis. One also has to examine the supplementary file 4 for details of frameworks used in the included process evaluations, and little detail is extended to this point within the discussion. Again, given the title,	Thank you to the reviewer for these comments. In relation to the first point, we have included content about the interventions in supplementary file 5 and focused on the interventions in the first part of our narrative synthesis titled 'descriptions of interventions and their causal assumptions.' This content aims to address both the first and second objectives and the newly developed figure 2 outlines how our objectives align with the MRC framework. To address the second point made here about the quality of the process evaluations, we have added sentences to the 'included process evaluation' section on lines 310-317, page 10. Within this paragraph we have also included examples of other frameworks that are outlined in supplementary file 4 as requested. In the 'strengths and limitations' part of the discussion section, we have extended the section where we commented on the quality of process evaluations to consider how including frameworks might impact on quality. We have also added text here about the importance of ensuring some consistency in how frameworks are used and reported. In this section we have also referred to the new MRC guidance that was recently published since this review was submitted to emphasise the importance of programme theories and logic models and theoretical understandings of interventions as a basis for planning process evaluations.

and what may prompt one to read this article, prioritisation and expansion of this point, would in my opinion improve this article.	
Within strengths and limitations, where you make particular reference to age, I think it is also relevant to acknowledge the characteristic heterogeneity of populations such as those with intellectual disability and those with mental illness where clinical environments may exert strong influence on interventional engagement.	Thank you, the reviewer makes an important point in relation to how context can influence intervention engagement. However the studies where participants had mental illnesses (Williams 2019) or intellectual disabilities (Matthews et al., 2016) both include community based walking interventions called 'Walk Well' and 'Walk this Way.' Within these interventions there is no clinical environment associated with the participants' health conditions/characteristics that might influence their engagement. Based on this, we have decided not to include the reviewer's suggestion in the discussion section.
The authors should mention in their discussion other implementation outcomes that may be relevant, but less explored through their framework but nonetheless, potentially relevant, such as outlined in the work of Proctor et al. 2011.	Thank you to the reviewer for highlighting the work of Proctor et al. (2011). We have incorporated this into the discussion section (page 19, lines 730-739).
Among included studies, Suppl file 3, Berendsen (2015): Can the authors clarify that the control conditions meet the inclusion criteria and do not extend to exclusionary. I didn't understand the nature of control conditions.	According to the published protocol (Corepal et al., 2019), eligible comparators includes 'alternative treatments.' The selection of included studies followed this criterion, and thus Berendsen (2015), which compared two different treatments, was included in this review. We have now added this criterion into the "Comparator" section (page 4, lines 145-146), as per the protocol to aid clarification.
There is perhaps a comment needed within the discussion about the 'intense' nature of some of the control conditions, in the context of limited efficacy of SB interventions.	We would like to thank the reviewer for this interesting point. We have revised the following part of the discussion to include this content (page 17, lines 649-653).
Some editing work required on spacing, punctuation and consistency of capital lettering throughout.	We have checked the revised manuscript for formatting issues and corrected them throughout

VERSION 2 – REVIEW

REVIEWER	Craig, Kelly IBM Watson Health, Center for AI, Research, and Evaluation Employed by IBM Corporation
REVIEW RETURNED	29-Nov-2021
GENERAL COMMENTS	There are some very minor spelling and formatting issues.

	Also in line 343 of track changes check reporting: you listed black/native American, presumably, the slash should be a comma. "Only nine trials reported ethnicity, the most ethnically diverse study was by Albright et al., (28) which reported the following ethnicities: Native Hawaiian, Pacific Islander, Asian, mixed race, white, black/native American." Note, "native" should be capitalized.
--	--